# Effective Utilization of *Vaccinium virgatum* Aiton Stems as Functional Materials: Major Constituent Analysis and Bioactivity Evaluation

**DOI:** 10.3390/plants11040568

**Published:** 2022-02-21

**Authors:** Hisahiro Kai, Kazuhiro Sugamoto, Saki Toshima, Yo Goto, Takayuki Nakayama, Kazuhiro Morishita, Hisato Kunitake

**Affiliations:** 1Department of Pharmaceutical Health Sciences, School of Pharmaceutical Sciences, Kyushu University of Health and Welfare, Nobeoka 882-8508, Japan; 2Department of Applied Chemistry, Faculty of Engineering, University of Miyazaki, Miyazaki 889-2192, Japan; sugamoto@cc.miyazaki-u.ac.jp; 3Department of Biochemistry and Applied Biosciences, Faculty of Agriculture, University of Miyazaki, Miyazaki 889-2192, Japan; na20006@student.miyazaki-u.ac.jp (S.T.); hkuni@cc.miyazaki-u.ac.jp (H.K.); 4Biolabo Co., Ltd., Kobe 650-0047, Japan; ygoto@ge-hd.co.jp (Y.G.); tnakayama@ge-hd.co.jp (T.N.); 5Division of Tumor and Cellular Biochemistry, Department of Medical Sciences, Faculty of Medicine, University of Miyazaki, Miyazaki 889-2192, Japan; kmorishi@med.miyazaki-u.ac.jp

**Keywords:** *Vaccinium virgatum*, stem, proanthocyanidin (PAC), antioxidant activity, adult T cell leukemia (ATL)

## Abstract

We previously reported that rabbit-eye blueberry (*Vaccinium virgatum* Aiton) leaves exhibit multiple functions. In this study, we evaluated whether *V. virgatum* stems can also be used as functional materials similar to leaves and clarified the major constituents and their biological activity (antioxidant activity and anti–adult T cell leukemia (ATL) activity). Water extracts of *V. virgatum* stems were separated into 19 fractions using a Diaion HP-20 open column. Sugars and organic acids were detected in the highly water-soluble fractions. Polyphenols and proanthocyanidin were detected in the hydrous methanol-soluble fractions. In biological activity evaluations, a difference in antioxidant activity was observed in the water-containing methanol-eluted fractions, and fractions exhibiting anti-ATL activity differed depending on cell type. These results suggest that blueberry stems, like leaves, are rich in polyphenols and exhibit antioxidant activity and inhibit ATL cell growth. In the future, aerial parts of blueberries, including stems and leaves, could be used as functional materials and/or medicinal resources.

## 1. Introduction

Blueberry belongs to the Ericaceae family, and its fruits are eaten worldwide. Blueberry leaves have been used as an anti-diabetes folk medicine in Europe and Canada [1,2]. Although the fruits of the northern highbush blueberry (*Vaccinium corymbosum* L.) and southern highbush blueberry (*V. darrowii* Camp.) are often used for food, our research to date indicates that the functionality of rabbit-eye blueberry (*V. virgatum* Aiton), the fruit of which is generally not edible, resides primarily in the leaves. Although the raw fruit of *V. virgatum* is not eaten, the plant is cultivated because it is resistant to high temperatures and dryness and exhibits wide soil adaptability (pH 4.3–5.3) [3].

In a previous study that screened samples of 52 agricultural plants for the ability to inhibit the proliferation of seven adult T-cell leukemia (ATL)-related cell lines, blueberry leaves exhibited significant inhibitory effects [4]. ATL is a peripheral T-cell lymphoma caused by human T-cell leukemia virus type 1 (HTLV-1) [5]. Leaves of *V. virgatum* plants collected in December exhibited the most effective inhibition of ATL cell proliferation [6,7]. The leaves of *V. virgatum* plants also exhibit a variety of other functions, including effects on alcohol metabolism during chronic intake of ethanol (EtOH) [8], anti-fibrogenic properties [9], and suppression of hepatitis C virus replication [10]. Moreover, we reported a relationship between seasonal variation and ATL-related bioactivity of natural products using a direct-injection electron ionization–mass spectrometry metabolomics method [11].

Blueberry leaves appear to have considerable bioactivity as functional foods. Wu et al. reported on the total phenolic, total flavonoid, and proanthocyanidin contents in the leaf extracts from 73 different blueberry cultivars [12]. Tetsumura et al. reported that a new blueberry species of *V. virgatum* “Kunisato 35 Gou” had been established for leaf cultivation. The leaf of “Kunisato 35 Gou” enables the production of large amounts of polyphenols. “Kunisato 35 Gou” is cultivated in Miyazaki prefecture in the southern part of Japan [13]. We are currently evaluating the beneficial function of this species. However, the bioactivity of blueberry stems, which form part of the same nutritional organ, has not yet been clarified. The stems are not primarily used, however, so after leaves are harvested, almost all of the remaining plant material is discarded. As a result, all of the components contained in blueberry stems have not yet been clarified. In the present study, we investigated the characteristics of the components of *V. virgatum* stems and the biological activity (antioxidant activity and anti-ATL activity) of each extract and fraction.

## 2. Results

### 2.1. Fractionation of Water Extracts of Blueberry (V. virgatum) Stems

As blueberry stems contain numerous compounds of varying molecular size, we fractionated stem extracts by molecular size using gel filtration chromatography. Water extracts of *V. virgatum* stems were separated into 19 fractions using a Diaion HP-20 open column eluted with hydrous methanol and methanol. The yield of each extract fraction and elution solvent are shown in Table 1. As the plant material was extracted using water, the yield of the fractions eluted with water tended to be higher under these chromatography conditions than that of fraction eluted with hydrous methanol or methanol alone.

### 2.2. Sugar and Organic Acid Analyses

The potentially significant effect of sugars and organic acids on taste must be considered when using blueberry stems as a functional material. The amounts of various sugars and organic acids contained in blueberry stems and fruits were measured. The sugars analyzed included glucose, fructose, and sucrose, whereas the organic acids measured included quinic acid, citric acid, and malic acid. As shown in Table 2, sugars were detected only in the highly water-soluble fractions 2–5. As shown in Table 3, the organic acids, quinic acid, citric acid, and malic acid, were also detected in the highly water-soluble fractions 2–5. Quinic acid, citric acid, and malic acid were also detected in fraction 8, but the levels were not as high as in the other fractions. Organic acids were detected in more fractions than were sugars.

### 2.3. Polyphenol Analysis

Blueberry fruits are rich in bioactive polyphenols [14]. Previously, we investigated bioactive polyphenols in blueberry leaves, which are rarely used as foods [15,16], and in the present study, we analyzed polyphenols in stems as well as leaves. The total polyphenol content in the 19 fractions obtained from chromatographic separation of water extracts of *V. virgatum* stems according to the Folin–Ciocalteu reagent method is summarized in Figure 1. Fractions 10–14 exceeded 800 mg/g (gallic acid/fraction dry weight). From these results it can be seen that as the ratio of methanol in the elution solvent increased to >50%, the amount of polyphenol eluted increased markedly.

Following the total polyphenol analysis, we analyzed five major polyphenols (chlorogenic acid, catechin, epicatechin, rutin, and caffeic acid) individually using our previous method [17,18]. As shown in Table 4, of the 19 fractions, the highest amount of chlorogenic acid was found in fraction 6 (94.1 ± 6.4 mg/g), whereas the highest amount of catechin was found in fraction 10 (64.1 ± 35.2 mg/g), the highest amount of epicatechin in fraction 13 (73.7 ± 2.6 mg/g), the highest amount of rutin in fraction 15 (56.2 ± 19.2 mg/g), and the highest amount of caffeic acid in fraction 14 (18.8 ± 1.2 mg/g). As shown by these results, each of the major polyphenol components eluted in different fractions, and all of the polyphenols examined were eluted with hydrous methanol.

### 2.4. Proanthocyanidin (PAC) Analysis

Proanthocyanidins (PACs) are condensed tannins that exhibit varied pharmacological properties, and *V. virgatum* leaves are reportedly rich in PACs [19]. We therefore analyzed PACs in both blueberry stems and leaves. Figure 2 shows the PAC content of the 19 fractions obtained from water extracts of *V. virgatum* stems analyzed using the 4-dimethylaminocinnamaldehyde (DMACA) method. The PAC content of fractions 8–11 exceeded 200 mg in terms of catechin, and these fractions contained significantly more catechin than the remaining fractions.

### 2.5. Antioxidant Activity

A number of reports have shown that polyphenol-rich natural resources exhibit strong antioxidant activity [20]. Trolox is an antioxidant compound that is used in 1,1-diphenyl-2-picrylhydrazyl (DPPH) free radical scavenging assays. As shown in Figure 3, fractions 7–15 obtained from water extracts of *V. virgatum* stems exhibited stronger antioxidant activity than the remaining fractions. As shown in Figure 4, there was a positive correlation between the amount of PAC and antioxidant activity. The results of this analysis showed that fractions 8–11 were located at high levels of both PAC contents and antioxidant activity. This result suggests that the polyphenols and PACs eluted from the extract with hydrous methanol are the primary contributors to the antioxidant effect.

### 2.6. Inhibition of ATL-Related Cell Growth

In the anti-ATL activity bioassay, we analyzed 16 fractions obtained from the water extract of *V. virgatum* stems for the ability to inhibit the proliferation of four leukemia cell lines (two ATL cell lines, one human T-cell line transformed via HTLV-1 infection, and one HTLV-1–negative human T-cell acute lymphoblastic leukemia cell line) to identify fractions containing compounds active in the prevention of ATL. Only 16 fractions were analyzed because the amounts of fractions 1, 18, and 19 were insufficient for the experiment. Figure 5 shows the inhibitory effect of each fraction at 50 μg/mL on ATL-related cell proliferation. Genistein has inhibitory effects on ATL cells in vitro and in vivo, and was used as a positive control [21]. All fractions significantly inhibited the proliferation of ED and S1T ATL cells (Figure 5A,B). In addition to their antioxidant activity, fractions 7–15 exhibited a 50% growth inhibition compared with the control. Against HTLV-1–infected cells (MT-2), fractions 9–12 exhibited a 50% growth rate reduction compared with the control. Fractions 10 and 11 inhibited cell growth to 10% of the control (Figure 5C). Against the HTLV-1–negative human T-cell acute lymphoblastic leukemia cell line (Jurkat), fractions 10–14 inhibited growth to 50% of the control. Fractions 10–12 inhibited growth to approximately 20% of the control (Figure 5D). Interestingly, the extent of cell proliferation inhibition depended on the cell type; for fractions 7–9, this dependence on ATL cell type was particularly notable. The 50% inhibitory concentration (IC_50_) values for *V. virgatum* stem extract against S1T, ED, MT-2, and Jurkat cells were 64.5, 47.0, 103.8, and 126.3 μg/mL, respectively.

## 3. Discussion

This study evaluated the potential for using blueberry stems as functional materials, as this part of the plant is typically discarded as waste. A crude water extract of stems was prepared and fractionated by column chromatography, and the resulting fractions were analyzed for sugar, organic acid, and polyphenol content and bioactivity. Consistent with analyses of components of samples roughly fractionated by reverse-phase chromatography using the porous resin Diaion HP-20, sugars and organic acids eluted primarily in fractions eluted with only water, whereas polyphenols eluted with hydrous methanol or methanol. PACs eluted primarily with hydrous methanol fractions, but the amount eluting with low-polarity solvents such as methanol, acetone, and ethyl acetate was small. With regard to biological activity, the antioxidant activity varied across the water-containing methanol-eluted fractions, whereas the anti-ATL cell proliferation activity in fractions varied by cell type.

Crude drugs are primarily processed parts of plants that are said to have medicinal properties, such as leaves, stems, and roots. Such drugs have been identified based on many years of experience. It is important to use the appropriate specific part of a medicinal plant to prevent or treat a disease. However, for diseases such as ATL, for which no therapeutic drug has been established, there is ample opportunity to consider how to utilize previously overlooked natural resources. For example, we found the leaves and stems of cucumber (*Cucumis sativus*) are suitable natural resources for cosmetics. Cucumber leaves and stems are not generally used after the harvesting of the cucumber fruits, and therefore almost all of the remaining plant material becomes waste [22]. As mentioned in the Introduction, our group has found that blueberry leaves exhibit versatile functionality [4,5,6,7,8,9,10]. There are no reports comparing the leaves and fruits of the same individual, or similar plant species, related to the results of *V. virgatum* stem studies discussed in this paper. This is the first report characterizing the biological activity of potentially medicinal components contained in blueberry stems. In the future, we would like to further confirm the superiority of the blueberry waste disposal site by comparing the composition of leaves, stems, and fruits with the biological activity of the same individual.

## 4. Materials and Methods

### 4.1. Plant Material

Stems of *V. virgatum* (Kunisato 35 Gou) were collected in November and December 2017 in Aya, Miyazaki, Japan. The voucher specimen (180216) was deposited at Biolabo Co., Ltd. (Kobe, Hyogo, Japan).

### 4.2. Extraction and Fractionation

Prior to extraction, the samples were blanched. A steamer for tea leaves was used to steam stems of *V. virgatum* to inactivate enzymes. The stems were then partially dried by applying warm air, and after removing from rough heat, stems were stored in a freezer. Semi-dried *V. virgatum* stems (3000 kg) were extracted using H_2_O (7000 L for 90 min, 1 time) at 90–95 °C. The resulting H_2_O extract was lyophilized to yield 114.5 kg of residue. The lyophilized H_2_O extract (500 g) was subjected to column chromatography on a Diaion HP-20 column eluted stepwise using H_2_O:MeOH (1:0, 3:1, 1:1, 1:3, 0:1), acetone, and EtOAc (each 4L) to yield 19 corresponding fractions. The eluting solvent and yield of each fraction are shown Table 1.

### 4.3. Sugar and Organic Acid Analysis

Sugar and organic acid contents were determined by high-performance liquid chromatography (HPLC), according to the method of Toshima et al. [23]. Each fraction (0.02 g) was dissolved in 5 mL of ultrapure water and passed through a 0.22-μm membrane filter (Millipore, Bedford, MA, USA) prior to analysis. For sugar content analysis, the extracts were subjected to HPLC using a UF-Amino station system (Shimazu, Kyoto, Japan) equipped with a refractive index detector (RID-10A, Shimadzu) and Asahipak NH2P-50 4E column (Showadenko, Tokyo, Japan). The chromatographic conditions were as follows: solvent, 75% (*v*/*v*) acetonitrile (CH_3_CN); column temperature, 40 °C; flow rate, 1.0 mL/min. Retention times and spectra were compared with those of pure standards of glucose, fructose, and sucrose. For organic acid content determination, each fraction was analyzed by reverse-phase HPLC using a Prominence LC solution system (Shimadzu) equipped with a photodiode array detector (SPD-M20A, Shimadzu) and Inertsil ODS3 column (Shimadzu). The chromatographic conditions were as follows: solvent, 75% (*v*/*v*) CH_3_CN; column temperature 40 °C; flow rate, 1.2 mL/min. Retention times and spectra were compared with those of pure standards of quinic acid, citric acid, and malic acid. Results are expressed as milligrams per gram dry weight of each fraction. Sample extracts were analyzed three times.

### 4.4. Polyphenol Analysis

The amount of polyphenols in each fraction obtained from the stems of *V. virgatum* was determined according to the method of Toyama et al. [18]. For polyphenol analysis, each fraction (0.02 g) was prepared by dissolving in 5 mL of 80% (*v*/*v*) MeOH and passing through a 0.22-μm membrane filter (Millipore). Total polyphenol content was determined according to the Folin–Ciocalteu reagent method [24]. Briefly, 200 μL of each sample, 200 μL of phenol reagent (distilled water (5.0 mL) and Folin–Ciocalteu’s phenol reagent), and 400 μL of saturated sodium carbonate solution were mixed. Absorbance was read at 760 nm after 30 min. Gallic acid dissolved in 80% (*v*/*v*) MeOH was used as a standard. Total polyphenol content was expressed as grams of gallic acid equivalent/100 g of each fraction. Each sample was measured three times.

The polyphenol components of each fraction were identified by HPLC. For sample preparation, 0.02 g of each fraction was dissolved in 5 mL of 80% (*v*/*v*) MeOH. The solution was passed through a 0.22-μm membrane filter (Millipore) for HPLC analysis, which was performed using a Prominence LC solution system (Shimadzu) equipped with an Inertsil ODS3 column (Shimadzu; 4.6 mm × 250 mm, 5 μm). The chromatographic conditions were as follows: solvent A, 100% (*v*/*v*) EtOH; solvent B, 20 mM potassium phosphate (pH 2.4); column temperature, 40 °C; detection at 280 nm; flow rate, 1.0 mL/min. The binary gradient was as follows: 85–68% B (0–12 min), 68% B (12–15 min), 68–55% B (15–20 min), 55–85% B (20 min), 85% B (20–29 min). Retention times and spectra were compared with pure standards of chlorogenic acid, catechin, epicatechin, rutin, and caffeic acid. These extracts were diluted within the range 20 to 200 mg/L using 100% (*v*/*v*) MeOH. The concentrations were designed to yield comparable peak heights for easy derivation of chromatographic parameters. The peak areas of the standard and samples were normalized and used for the quantitation of active components, the contents of which were subsequently expressed as percentage of the label claim. The results were expressed as grams of gallic acid equivalent (mg)/100 g of each fraction. Each sample was measured three times.

### 4.5. PAC Analysis

PAC content was determined according to the DMACA–HCl protocol using a modification of the method of Li et al. [25]. A total of 25 mg of each fraction was dissolved in 5 mL of ultrapure water and passed through a 0.22-μm membrane filter (Millipore, Burlington, MA, US) prior to analysis. Next, 30 μL of sample or catechin standard solution and 170 μL of DMACA solution (0.1%, dissolved in 1 M HCl-MeOH solvent) were mixed. The amount of DMACA sample (or catechin) complex in the solution was determined by measurement of the optical density at 640 nm after incubating 20 min at room temperature. The standard curve was linear between 2 and 40 μg/mL of catechin. Results are expressed as substantial quantities of catechin content/fraction dry weight (mg/g). Each sample was measured three times.

### 4.6. Antioxidant Activity

Antioxidant activity was measured according to the method of Toyama et al. [18]. Each sample was prepared by dissolving freeze-dried fruit powder (0.1 g) in 80% (*v*/*v*) EtOH and passing through a 0.22-μm membrane filter (Millipore). Antioxidant activity was determined using a DPPH free radical scavenging assay. Fifty microliters of 20% EtOH and 50 μL of 200 mM 2-morpholinoethanesulphonic acid buffer (pH 6.0) were added to 50 μL of sample solution per well in a 96-well microplate. The reaction was initiated by the addition of 50 μL of 1.2 mM DPPH to EtOH. Twenty minutes after adding the reagents and incubating at room temperature, the absorbance at 520 nm was measured using an Immuno-Mini NJ-2300 microplate reader (Nalge Nunc International, Tokyo, Japan), with Trolox (Aldrich Japan, Tokyo, Japan) used as a standard. Antioxidant activity is expressed as μmol Trolox equivalents/g of each fraction. Each sample extract was measured three times. The standard curve was linear between 0 and 50 ppm Trolox. Results are expressed as substantial quantities of trolox content/fraction dry weight (μmol/g). Each sample was measured three times.

### 4.7. ATL Assay

ED cells were kindly provided by Dr M. Maeda (Kyoto University, Japan), and S1T cells were kindly provided by Dr N. Arima (Kagoshima University, Japan). MT-2 and Jurkat cells were obtained from the Fujisaki Cell Center, Hayashibara Biochemical Laboratories (Okayama, Japan) [26]. All cells were maintained in RPMI 1640 medium (Sigma-Aldrich Co., St. Louis, MO, USA) supplemented with 10% fetal bovine serum (Sigma-Aldrich Co., lot no. 13C491) containing 100 U/mL penicillin G and 100 μg/mL streptomycin (Life Technologies, Carlsbad, CA, USA). Each cell line was seeded (1 × 10^5^ cells/mL, 90 μL/well) into a 96-well plate containing RPMI 1640 medium. After incubation at 37 °C for 24 h in an atmosphere containing 5% CO_2_, samples were added (10 μL/well) to the cells and incubated for an additional 72 h. Subsequently, the inhibition of cell proliferation was determined using a 2-(2-methoxy-4-nitrophenyl)-3-(4-nitrophenyl)-5-(2,4-disulfophenyl)-2H-tetrazolium monosodium salt (WST-8) assay kit (Dojindo, Kumamoto, Japan). Viable cells convert the tetrazolium salt in WST-8 to highly water-soluble formazan, which is monitored by measuring the absorbance at 450 nm using a microplate reader (Multiskan FC, Thermo Fisher Scientific, Waltham, MA, USA). Genistein (Wako, Osaka, Japan) was used as a positive control [21].

### 4.8. Statistical Analysis

All experimental results were obtained from triplicate measurements (three extracts and three measurements per extract), and the data in the tables and figures represent mean values ± standard deviation (*n* = 3). Differences were evaluated for statistical significance at *p* = 0.05 using univariate analysis of variance with Statistica 13.0 software (StatSoft, Tulsa, OK, USA) and Tukey’s post hoc test. We investigated the correlations among total anthocyanin content, total polyphenol content, antioxidant activity (DPPH), and anti–cancer cell proliferation properties.

## 5. Conclusions

Blueberry stems, like leaves, are rich in polyphenols and exhibit antioxidant and ATL cell growth inhibitory activity. In the future, blueberries are expected to be used as functional materials and/or medicinal resources as aerial parts, including both stems and leaves.

## Figures and Tables

**Figure 1 plants-11-00568-f001:**
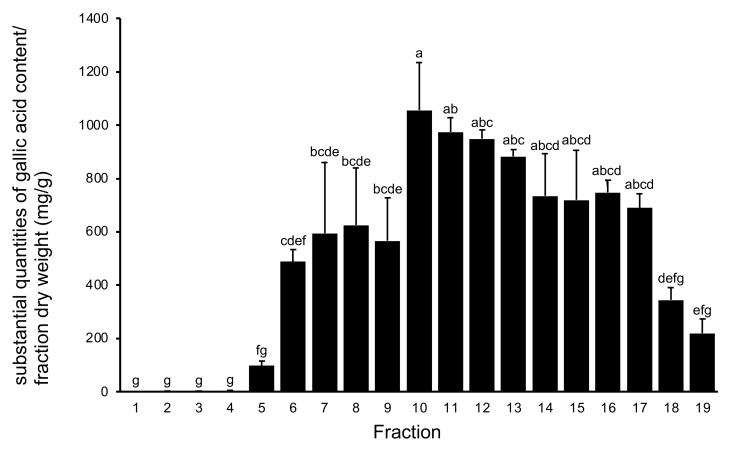
Total polyphenol content of fractions from water extracts of *V. virgatum* stems. (mg/g (dry weight of fraction)). Data are mean ± SD for three independent experiments. Different letters represent significant differences by Tukey’s multiple comparison test (*p* = 0.01).

**Figure 2 plants-11-00568-f002:**
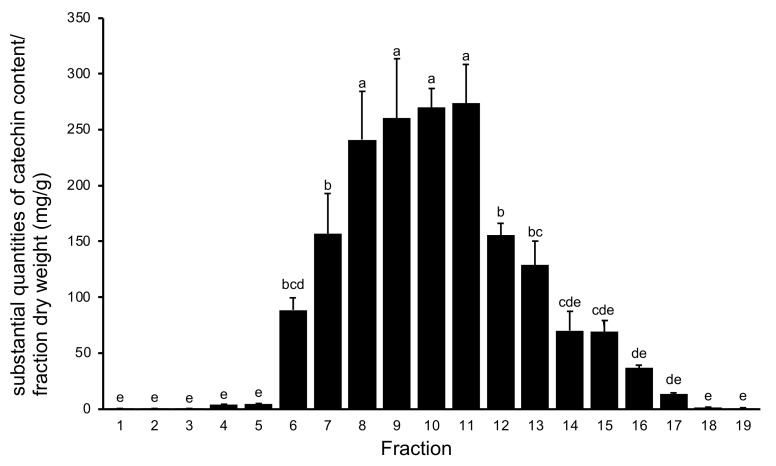
Proanthocyanidin content of fractions from water extracts of *V. virgatum* stems. Data are mean ± SD for three independent experiments. Different letters represent significant differences by Tukey’s multiple comparison test (*p* = 0.01).

**Figure 3 plants-11-00568-f003:**
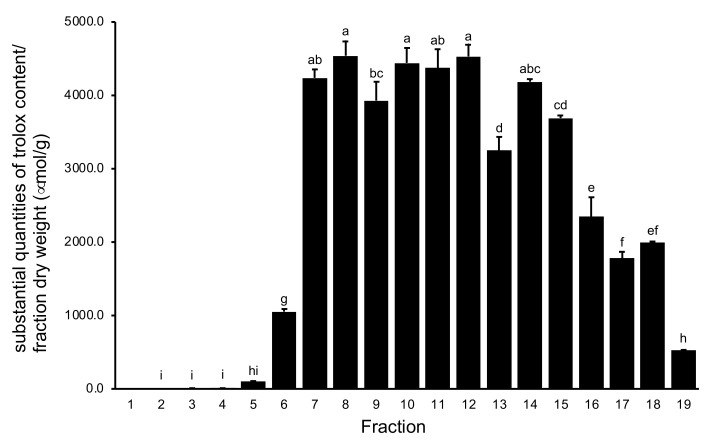
Antioxidant activities of Fractions from Water Extracts of *V. virgatum* stems. Data are mean ± SD for three independent experiments. Different letters represent significant differences by Tukey’s multiple comparison test (*p* = 0.01). Fraction 1 was not detected.

**Figure 4 plants-11-00568-f004:**
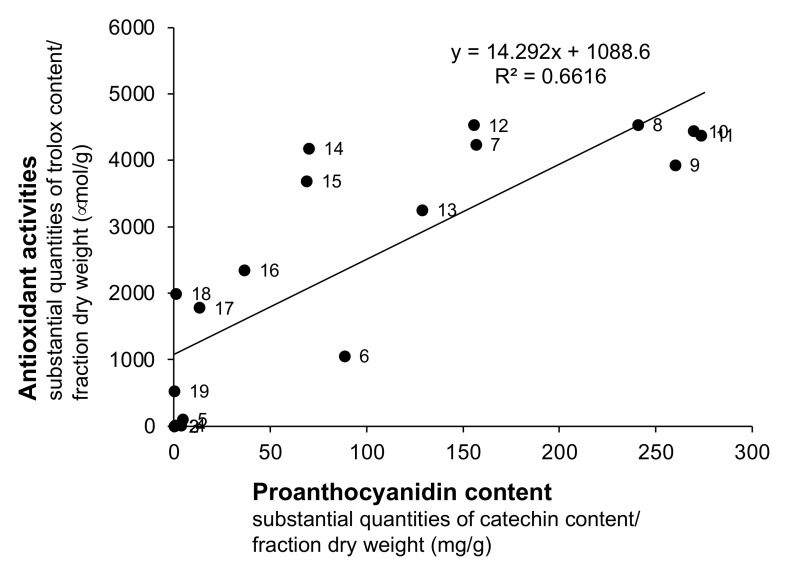
Correlation between Proanthocyanidin content and Antioxidant activities of Fractions from Water Extracts of *V. virgatum* stems. The label on each plot indicates the fraction number. Fraction 1 cannot be displayed because it could not be detected by the antioxidant activities.

**Figure 5 plants-11-00568-f005:**
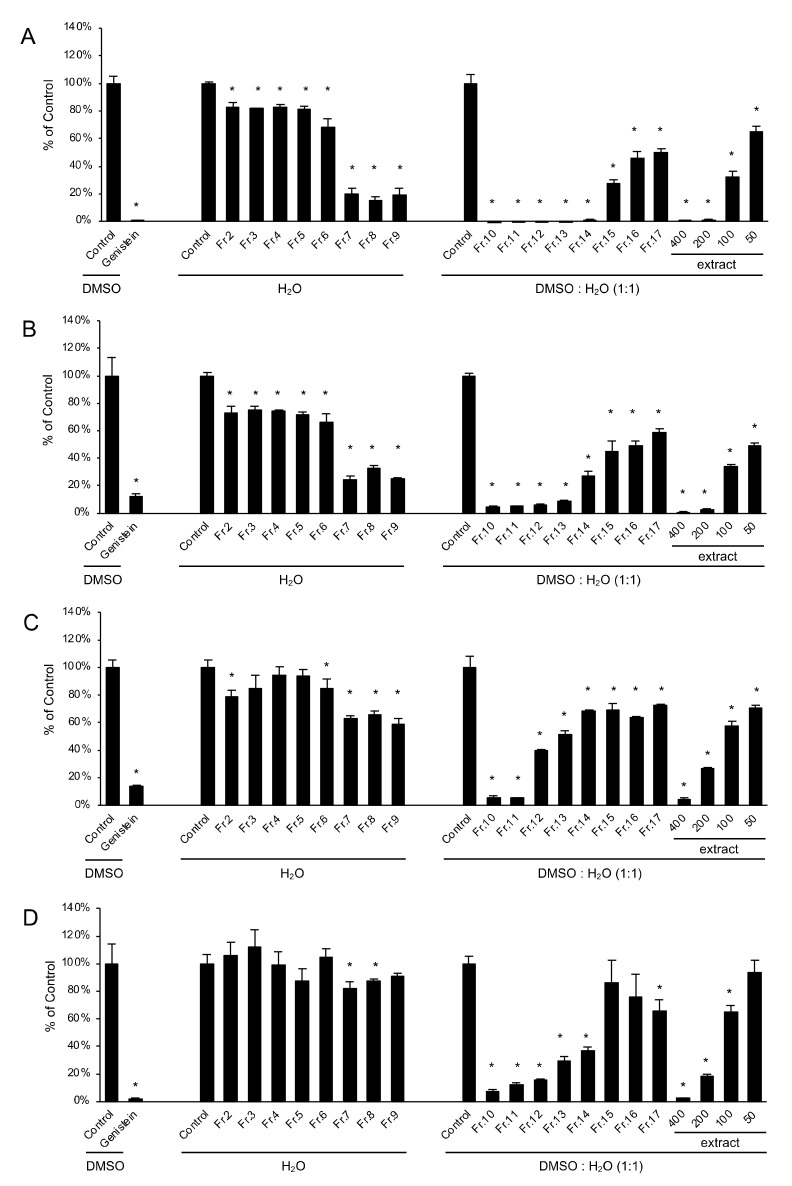
Inhibitory effects against growth of ATL cell lines and antioxidant activity of fractions from water extracts of *V. virgatum* stems. (**A**) S1T cell, (**B**) ED cell, (**C**) MT-2 cell, (**D**) Jurkat cell. S1T and ED were ATL cell lines, MT-2 was HTLV-1–infected cell line and Jurkat HTLV-1–negative human T-cell acute lymphoblastic leukemia cell line. Concentration of all fractions and genistein were 50 μg/mL. Concentrations of blueberry stem extract were 400, 200, 100, and 50 μg/mL. Data are mean ± SD for three independent experiments. Significantly different from each control value: *p* < 0.05 (*) (Student’s *t*-test). The amounts of fractions 1, 18, and 19 were insufficient for the experiment. Considering the polarity of each fraction, fractions 2–9 were dissolved in H_2_O, and fractions 10–17 and extracts were dissolved in DMSO:H_2_O (1:1). Genistein was used as a positive control and dissolved in DMSO. These solutions were then diluted with medium to a final concentration of 0.1% for the first solvent (H_2_O or DMSO:H_2_O (1:1) or DMSO) used.

**Table 1 plants-11-00568-t001:** Yield of fractions of *V. virgatum* stem extract separated using HP-20 column chromatography.

Fraction	Eluting Solvent	Fraction Yield (g)
1	H_2_O	0.5
2	H_2_O	76.7
3	H_2_O	190.4
4	H_2_O	144.8
5	H_2_O:MeOH (3:1)	12.7
6	H_2_O:MeOH (3:1)	5.8
7	H_2_O:MeOH (3:1)	13.2
8	H_2_O:MeOH (3:1)	1.5
9	H_2_O:MeOH (2:2)	2.2
10	H_2_O:MeOH (2:2)	29.1
11	H_2_O:MeOH (2:2)	25.6
12	H_2_O:MeOH (2:2)	3.7
13	H_2_O:MeOH (1:3)	4.8
14	H_2_O:MeOH (1:3)	2.7
15	H_2_O:MeOH (1:3)	1.1
16	MeOH	4.5
17	MeOH	3.2
18	MeOH	0.8
19	Acetone, EtOAc *	0.7

The lyophilized H_2_O extract (500 g) was subjected to column chromatography on a Diaion HP-20 column eluted stepwise using H_2_O:MeOH (1:0, 3:1, 1:1, 1:3, 0:1), acetone, and EtOAc (each 4 L) to yield 19 corresponding fractions. * EtOAc: ethyl acetate.

**Table 2 plants-11-00568-t002:** Sugar content of fractions from water extracts of *V. virgatum* stems (mg/g (dry weight of fraction)).

Fraction	Glucose	Fructose	Sucrose
1 (H_2_O)	n.d.	n.d.	n.d.
2 (H_2_O)	277.0 ± 8.3	267.0 ± 14.1	79.2 ± 9.1
3 (H_2_O)	279.2 ± 28.5	261.0 ± 30.6	75.7 ± 1.7
4 (H_2_O)	242.2 ± 16.1	217.8 ± 8.6	66.9 ± 7.9
5 (H_2_O:MeOH = 3:1)	166.8 ± 11.0	147.3 ± 10.3	42.0 ± 7.0
6 (H_2_O:MeOH = 3:1)	n.d.	n.d.	n.d.
7 (H_2_O:MeOH = 3:1)	n.d.	n.d.	n.d.
8 (H_2_O:MeOH = 3:1)	n.d.	n.d.	n.d.
9 (H_2_O:MeOH = 1:1)	n.d.	n.d.	n.d.
10 (H_2_O:MeOH = 1:1)	n.d.	n.d.	n.d.
11 (H_2_O:MeOH = 1:1)	n.d.	n.d.	n.d.
12 (H_2_O:MeOH = 1:1)	n.d.	n.d.	n.d.
13 (H_2_O:MeOH = 1:3)	n.d.	n.d.	n.d.
14 (H_2_O:MeOH = 1:3)	n.d.	n.d.	n.d.
15 (H_2_O:MeOH = 1:3)	n.d.	n.d.	n.d.
16 (MeOH)	n.d.	n.d.	n.d.
17 (MeOH)	n.d.	n.d.	n.d.
18 (MeOH)	n.d.	n.d.	n.d.
19 (Acetone, EtOAc)	n.d.	n.d.	n.d.

Data are mean ± SD for three independent experiments: n.d., not detected.

**Table 3 plants-11-00568-t003:** Organic acid content of fractions from water extracts of *V. virgatum* stems (mg/g (dry weight of fraction)).

Fraction	Quinic Acid	Citric Acid	Malic Acid
1 (H_2_O)	4.1 ± 0.9	n.d.	2.2 ± 0.2
2 (H_2_O)	164.2 ± 13.7	25.4 ± 3.8	45.3 ± 9.0
3 (H_2_O)	118.8 ± 5.6	18.3 ± 0.9	36.0 ± 0.9
4 (H_2_O)	175.2 ± 6.9	26.4 ± 1.5	53.2 ± 4.5
5 (H_2_O:MeOH = 3:1)	131.1 ± 5.8	31.7 ± 1.2	67.8 ± 8.8
6 (H_2_O:MeOH = 3:1)	15.4 ± 4.6	n.d.	n.d.
7 (H_2_O:MeOH = 3:1)	n.d.	n.d.	n.d.
8 (H_2_O:MeOH = 3:1)	20.1 ± 9.7	24.5 ± 6.1	25.1 ± 7.8
9 (H_2_O:MeOH = 1:1)	n.d.	22.0 ± 3.0	n.d.
10 (H_2_O:MeOH = 1:1)	n.d.	2.5 ± 1.2	n.d.
11 (H_2_O:MeOH = 1:1)	n.d.	n.d.	n.d.
12 (H_2_O:MeOH = 1:1)	n.d.	n.d.	n.d.
13 (H_2_O:MeOH = 1:3)	n.d.	n.d.	n.d.
14 (H_2_O:MeOH = 1:3)	n.d.	n.d.	n.d.
15 (H_2_O:MeOH = 1:3)	n.d.	n.d.	n.d.
16 (MeOH)	n.d.	n.d.	n.d.
17 (MeOH)	n.d.	n.d.	n.d.
18 (MeOH)	n.d.	n.d.	n.d.
19 (Acetone, EtOAc)	n.d.	n.d.	n.d.

Data are mean ± SD for three independent experiments: n.d., not detected.

**Table 4 plants-11-00568-t004:** Content of major polyphenols in fractions from water extracts of *V. virgatum* stems (mg/g (dry weight of fraction)).

Fraction	Chlorogenic Acid	Catechin	Epicatechin	Rutin	Caffeic Acid
1 (H_2_O)	n.d.	n.d.	n.d.	n.d.	n.d.
2 (H_2_O)	n.d.	n.d.	n.d.	n.d.	n.d.
3 (H_2_O)	1.6 ± 0.1	n.d.	n.d.	n.d.	n.d.
4 (H_2_O)	3.2 ± 1.4	2.0 ± 0.5	n.d.	n.d.	n.d.
5 (H_2_O:MeOH = 3:1)	3.9 ± 1.9	3.3 ± 1.4	n.d.	n.d.	n.d.
6 (H_2_O:MeOH = 3:1)	94.1 ± 6.4	36.5 ± 2.0	n.d.	n.d.	13.2 ± 0.7
7 (H_2_O:MeOH = 3:1)	43.2 ± 9.1	26.0 ± 5.5	n.d.	n.d.	1.7 ± 0.1
8 (H_2_O:MeOH = 3:1)	36.1 ± 6.8	35.0 ± 6.6	13.1 ± 2.5	n.d.	1.7 ± 1.0
9 (H_2_O:MeOH = 1:1)	50.8 ± 23.6	60.6 ± 28.3	29.7 ± 14.6	n.d.	4.1 ± 1.5
10 (H_2_O:MeOH = 1:1)	22.3 ± 2.6	64.1 ± 35.2	25.8 ± 13.5	n.d.	13.6 ± 0.5
11 (H_2_O:MeOH = 1:1)	6.3 ± 2.1	36.9 ± 20.2	54.6 ± 36.5	2.0 ± 0.3	7.8 ± 2.5
12 (H_2_O:MeOH = 1:1)	3.2 ± 1.0	17.5 ± 8.2	59.6 ± 28.8	4.7 ± 1.7	10.9 ± 4.8
13 (H_2_O:MeOH = 1:3)	3.3 ± 0.1	18.9 ± 1.2	73.7 ± 2.6	12.2 ± 1.0	17.7 ± 0.9
14 (H_2_O:MeOH = 1:3)	11.6 ± 0.2	14.1 ± 0.3	11.8 ± 0.5	39.9 ± 1.4	18.8 ± 1.2
15 (H_2_O:MeOH = 1:3)	13.8 ± 0.2	16.1 ± 0.4	5.3 ± 0.7	56.2 ± 19.2	n.d.
16 (MeOH)	n.d.	14.9 ± 0.2	n.d.	27.9 ± 0.7	14.1 ± 0.3
17 (MeOH)	1.4 ± 0.0	5.7 ± 1.2	0.5 ± 0.1	13.8 ± 3.5	2.4 ± 0.3
18 (MeOH)	2.3 ± 0.2	3.9 ± 0.3	n.d.	4.7 ± 0.7	2.5 ± 0.1
19 (Acetone, EtOAc)	2.5 ± 0.2	3.4 ± 0.4	n.d.	n.d.	n.d.

Data are mean ± SD for three independent experiments: n.d., not detected.

## Data Availability

Not applicable.

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
