# Peer review of "Effective Utilization of Vaccinium virgatum Aiton Stems as Functional Materials: Major Constituent Analysis and Bioactivity Evaluation"

_plants, 2022, doi:10.3390/plants11040568_

Round 1
Reviewer 1 Report
The manuscript from Kai et al reports a straight forward analysis of the polar compounds found in the stems of the blueberry variety Vaccinium virgatum. The methods used appear to be thorough and the results correctly analysed. The text is clear and well written.
I have only a few questions and suggestions.
Table 1 lists the yield of the different fractions in grams but only in the Materials and Methods section is it specified how much was loaded. Including this number as a footnote would allow the reader to understand what proportion of the material was eluted in the different fractions.
Table 2 has many "n.d." entries. The footnote defines these as "not determined" but from the methods it appears that all fractions were tested for the presence of all compounds. If this is true then it would be better to label fractions where the compounds were not found as "not detected", rather than "not determined", to emphasise that the analysis was performed but the concentration was below the detection limit.
Section 2.5 states "This result suggests that the polyphenols and PACs eluted from the extract with hydrous methanol are the primary contributors to the antioxidant effect". If this is true, plotting the concentration of the polyphenols or PACs in the different fractions against the antioxidant activity one should see a correlation. Repeating this for the sugars and organic acids should not give a correlation. This would provide evidence for the statement.
I could not find a comparison of the content of the blueberry stems reported in this manuscript with what has been previously found in leaves or berries of this plant, or even of other blueberry species. This would add interesting context to the discussion and increase the significance of the report.
Author Response
Answers to Reviewer: 1
Point 1: Table 1 lists the yield of the different fractions in grams but only in the Materials and Methods section is it specified how much was loaded. Including this number as a footnote would allow the reader to understand what proportion of the material was eluted in the different fractions.
Response 1: Thank you for your good suggestion. According to your suggestion, we added content described in Materials and Methods section to the footnotes in the Table 1 (Page 3 line 83). “The lyophilized H2O extract (500 g) was subjected to column chromatography on a Diaion HP-20 column eluted stepwise using H2O:MeOH (1:0, 3:1, 1:1, 1:3, 0:1), acetone, and EtOAc (each 4 L) to yield 19 corresponding fractions.”
In addition, Reviewer 2 suggested adding the volume of solvent (each 4 L).
Point 2: Table 2 has many "n.d." entries. The footnote defines these as "not determined" but from the methods it appears that all fractions were tested for the presence of all compounds. If this is true then it would be better to label fractions where the compounds were not found as "not detected", rather than "not determined", to emphasise that the analysis was performed but the concentration was below the detection limit.
Response 2: Thank you for your suggestion. According to your suggestion, we corrected "not determined " to "not detected" the footnotes in the Table 2, 3 and 4 (Page 3 line 100, Page 4 line 105 and Page 5 line 133). Regarding this point, we forgot to write “Fraction 1 was not detected.”, so we added the footnote in the Figure 3 (Page 7, line 174).
Point 3: Section 2.5 states "This result suggests that the polyphenols and PACs eluted from the extract with hydrous methanol are the primary contributors to the antioxidant effect". If this is true, plotting the concentration of the polyphenols or PACs in the different fractions against the antioxidant activity one should see a correlation. Repeating this for the sugars and organic acids should not give a correlation. This would provide evidence for the statement.
Response 3: Thank you for your questions. According to your suggestion, we revised the below.
・New figure about correlation between Proanthocyanidin content and Antioxidant activities was added (Figure 4, Page 7, line 175).
・We also added below sentence to make the explanation clear about Figure 4. “As shown in Figure 4, there was a positive correlation between the amount of PAC and antioxidant activity. The results of this analysis showed that fractions 8-11 were located at high levels of both PAC contents and antioxidant activity.” (Page 6, line 155)
・We delete the part of sentence “rather than the sugars and organic acids eluted with water”. (Page 6, line 159)
Point 4: I could not find a comparison of the content of the blueberry stems reported in this manuscript with what has been previously found in leaves or berries of this plant, or even of other blueberry species. This would add interesting context to the discussion and increase the significance of the report.
Response 4: Thank you for your good comments. We received encouraging suggestions for emphasizing the novelty of our research results. We have added the following text to the discussion section.
(Page 10, line 250) There are no reports comparing the leaves and fruits of the same individual, or similar plant species, related to the results of V. virgatum stem studies discussed in this paper.
(Page 10, line 253) In the future, we would like to further confirm the superiority of the blueberry waste disposal site by comparing the composition of leaves, stems and fruits with the biological activity of the same individual.
Thank you very much for your kind suggestions.
Reviewer 2 Report
The manuscript is focusing on identification and quantification of phenolic acids and main sugars in main crude fractions of Vaccinium virgatum stems extract and the results of subsequent biological tests. After reviewing, my comments concerning the manuscript are general positive, but needs some minor revision prior to acceptance:
- Page 9, line 215/216 “using H2O (7,000 kg”…) should be 7.0 L and the amount of obtained extract most likely is 1.145 kg not 114.5 kg.
- It would be correct if volumes of eluting solvents are noted.
- Have the same tests been performed with the extract?
Author Response
Answers to Reviewer: 2
Point 1: Page 9, line 215/216 “using H2O (7,000 kg”…) should be 7.0 L and the amount of obtained extract most likely is 1.145 kg not 114.5 kg.
It would be correct if volumes of eluting solvents are noted.
Response 1: Thank you for your questions. The numbers in this sentence were correct. As stated at the beginning of this sentence, we used 3,000 kg of stems for extraction. However, as suggested by the reviewers to avoid misunderstanding, change the unit of water from “kg” to “L” (Page 9, line 224). The solvent (H2O) volume 7,000 L (= 7,000 kg) for extraction was correct. In addition, we added “each 4 L”, which was the volumes of solvent using for column chromatography of Diaion HP-20, to both the footnote of table 1 (Page 3, line 84) and the section 4.1 (Page 10, line 271).
Point 2: Have the same tests been performed with the extract?
Response 2: Yes, all component analyzes, antioxidant activity test and ATL assay have been repeated 3 times. This is already mentioned in the footnotes of tables or figures.
Thank you very much for your kind suggestions.
Reviewer 3 Report
These results suggest that blueberry stems, like leaves, are rich in polyphenols and exhibit antioxidant activity and inhibit leukemia T cells growth.
The work is well structured but the introduction should be improved and more bibliographic references added.
Author Response
Answers to Reviewer: 3
Point : The work is well structured but the introduction should be improved and more bibliographic references added.
Response : Thank you for your good suggestion. We have revised following points.
・(Page 2, line 44) The recently review about adult T-cell leukemia (ATL) research was quoted. (Reference no. [5])
・(Page 2, line 52) We have added the following text.
“Wu et al. reported that the total phenolic, total flavonoid, and proanthocyanidin contents in the leaf extracts from 73 different blueberry cultivars [12]. Tetsumura et al. reported that a new blueberry species of V. virgatum "Kunisato 35 Gou" was established for leaf cultivation. The leaf of "Kunisato 35 Gou" enable the production of large amounts of pol-yphenols. "Kunisato 35 Gou" is cultivated in Miyazaki prefecture in the southern part of Japan.[13] We are evaluating the beneficial function of this species.”
Thank you very much for your kind suggestions.